# Genetic Diversity and Conservation of *Bomarea ovallei* (Phil.) Ravenna: Microsatellite Markers Reveal Population Vulnerability in the Atacama Desert

**DOI:** 10.3390/plants14101468

**Published:** 2025-05-14

**Authors:** Valeska Rozas-Lazcano, Mariel Mamani-Gómez, Irina Rojas-Jopia, Mariana Arias-Aburto, Roberto Contreras-Díaz

**Affiliations:** 1Departamento de Medicina, Facultad de Medicina, Universidad de Atacama, Copiapó 1532000, Chile; valeska.rozas.22@alumnos.uda.cl (V.R.-L.); mariel.mamani.22@alumnos.uda.cl (M.M.-G.); irina.rojas.22@alumnos.uda.cl (I.R.-J.); 2Complejo Tecnológico de Aprendizaje, Universidad de Atacama, Copiapó 1532000, Chile; mariana.arias@uda.cl; 3Centro Regional de Investigación y Desarrollo Sustentable de Atacama (CRIDESAT), Universidad de Atacama, Copiapó 1532000, Chile

**Keywords:** genetic diversity, *Bomarea* genus, Flowering Desert

## Abstract

The Atacama Desert, the driest and oldest desert on Earth, hosts a unique floral phenomenon known as the Desierto Florido (Flowering Desert), which occurs sporadically in response to rare rainfall events. *Bomarea ovallei* (Phil.) Ravenna is an endemic and endangered species of the Atacama Desert. However, its populations are geographically restricted and potentially vulnerable to genetic erosion due to isolation and extreme environmental conditions. This study aims to assess the genetic diversity of *B. ovallei* populations and develop microsatellite markers using next-generation sequencing (NGS) technology. A total of 268 microsatellite loci were identified, and 34 co-dominant markers were successfully developed for the first time in *B. ovallei*. Genetic diversity analysis using eight fluorescently labeled SSR markers revealed low genetic diversity across four populations, with the highest diversity observed in the QCA population, located within Llanos de Challe National Park, and the lowest in the TOTO population, which is highly exposed to anthropogenic activities. UPGMA and STRUCTURE analyses revealed three genetic clusters and high admixture among populations, suggesting historical or ongoing gene flow despite geographical separation. The presence of non-polymorphic loci and low PIC values in some markers further supports limited genetic variation. The newly developed microsatellite markers offer a valuable tool for future genetic studies, enabling the monitoring of genetic diversity and informing strategies for the preservation of this rare and ecologically significant species.

## 1. Introduction

The Atacama Desert, the driest and oldest desert on Earth, began to form during the Miocene, approximately 10 to 15 million years ago, and became more extreme during the Pliocene [1]. In certain areas, the sporadic rains, often linked to the El Niño–Southern Oscillation (ENSO) phenomenon, have unveiled a botanical treasure of significant scientific interest and tourist appeal [2,3]. The flora of the Atacama Desert has adapted to endure prolonged droughts, with plant structures such as bulbs and seeds lying dormant for years until the rains trigger their growth and flowering [4]. This phenomenon, known as the “Desierto Florido” (Flowering Desert), occurs with remarkable intensity in the Atacama Region of Chile [2].

Among the most iconic and endangered endemic species of the Atacama’s Flowering Desert is *Bomarea ovallei* (Phil.) Ravenna (*Alstroemeriaceae*), commonly called “Garra de León” (Lion’s Claw) (Figure 1) [5]. The genus *Bomarea* comprises approximately 120 described species, distributed from Mexico to Argentina and Chile [6,7,8], and is divided into four subgenera: *Bomarea* s.str., *Baccata*, *Sphaerine*, and *Wichuraea* [7,9]. In Chile, there are four species of the genus *Bomarea*: *Bomarea involucrosa* (Herb.) Baker, *Bomarea dulcis* (Hook.) Beauverd, *Bomarea salsilla* (L.) Mirb., and *B. ovallei* [5,10]. While *B. involucrosa* and *B. dulcis* are found in the extreme north of Chile (18° S and 19° S), with additional populations in Bolivia, and *B. salsilla* extends into central-southern Chile (33° S to 40° S), *B. ovallei* is isolated in the Atacama Region (28° S) with very few populations [5].

The populations of *B. ovallei* are located from the coastal zone (sea level) to inland valleys (160 m above sea level), between the southern limit of Llanos de Challe National Park (28°10′) and the northern limit of the town of Totoral (28°50′). The isolation of this endemic species in the Atacama Region may be explained by the fact that, although the genus *Bomarea* was once widely distributed across South America, the vast and arid Atacama Desert led to the disappearance of several species [7]. The extreme environmental conditions likely hindered the dispersal of propagules, and small populations, such as those of *B. ovallei*, may have experienced increased genetic drift, leading to the fixation of mutations [7].

This raises critical questions: What is the genetic diversity of *B. ovallei* populations? Can genetic markers be developed to assess the current status of these populations? Studying the genetic diversity of this species is crucial from an ecological, evolutionary, and conservation perspective, particularly in light of the challenges posed by climate change.

DNA markers are indispensable tools for studying plant species, providing detailed insights into their genetic diversity, population structure, and evolutionary relationships, all of which are vital for understanding ecosystems [11,12,13]. Among the most widely used DNA markers in plant genetics are microsatellites, also known as simple sequence repeats (SSRs). SSRs are short, tandemly repeated DNA sequences composed of mononucleotide, dinucleotide, trinucleotide, tetranucleotide, pentanucleotide, or hexanucleotide motifs. SSR markers can be broadly classified into two main categories: conventional SSRs, which are typically visualized using silver staining after gel electrophoresis, and fluorescently labeled SSRs, which are analyzed using capillary electrophoresis platforms [14]. These sequences are highly abundant, polymorphic, and randomly dispersed throughout the nuclear, chloroplast, and mitochondrial genomes of many species [12,15]. SSRs are particularly valuable for studying genetic diversity, evolutionary relationships, and genome structure, as their high polymorphism allows for the comparison and analysis of specific loci to establish close relationships [12,15]. Furthermore, the advent of next-generation sequencing (NGS) platforms has made it more efficient and cost-effective to generate large amounts of sequence data containing SSR motifs [15,16].

Molecular markers can also be used to assess the level of inbreeding, which can reduce the fitness of plants and increase their risk of extinction [17], especially in small or isolated populations [18]. Given the isolated nature of *B. ovallei* populations, it would be highly relevant to develop de novo nuclear markers using next-generation sequencing (NGS) technology and microsatellite primers for this species. Notably, such markers have not yet been recorded or published for *B. ovallei* or any other species within the genus *Bomarea*.

We hypothesize that the genetic diversity of *B. ovallei* populations is low, due to their geographic isolation and the challenges of seed reproduction and dispersal in the extreme conditions of the Atacama Desert. Therefore, our aim is to identify neutral markers using NGS technology in *B. ovallei* to analyze the genetic diversity of its populations. This research will provide critical insights into the conservation and evolutionary dynamics of this unique and endangered species.

## 2. Results

A total of 268 microsatellite loci were identified de novo from the assembled contigs generated via next-generation sequencing (NGS), utilizing the MegaSSR 2025 software (Table 1). Based on the SSR motif search parameters, trinucleotide repeats were the most abundant, accounting for 137 (51.11%) of the total SSRs. These were followed by dinucleotide repeats (99; 36.94%), tetranucleotide repeats (12; 4.47%), hexanucleotide repeats (10; 3.73%), mononucleotide repeats (seven; 2.61%), and pentanucleotide repeats (three; 1.11%) (Figure 2).

In terms of the distribution of microsatellites by motif type, TTA trinucleotide repeats were the most highly represented, with 13 occurrences in *B. ovallei* sequences. In contrast, motifs such as CTC, GGT, CGT, GCA, CTG, TGT, and ACG were less frequent, each appearing only once (Figure 3). Among the dinucleotide tandem repeats, the TC dimer was the most frequent, with 22 occurrences, followed by TA (17), AT (16), GA (14), and CT (11) dimers. For tetranucleotide motifs, the AATT repeat was the most common, appearing three times, while motifs such as TTTA and ATTA were less represented (Figure 3).

A set of 34 primer pairs was randomly selected to validate SSR locus amplification (Appendix A). These primer pairs were designed from DNA sequences containing trinucleotide motifs (AGA, TTA, TTC, TCA, TGA, TTG, and AAC) and dinucleotide motifs (TA, AG, and CT (Appendix A). The size range of the PCR products generated by these primer pairs varied between 103 and 287 bp (Appendix A). From these, a subset of eight primer pairs were chosen for fragment analysis to determine allele sizes in 39 individuals of *B. ovallei* from four populations (Table 2).

A total of 24 alleles were detected among *B. ovallei* individuals. The number of alleles generated by each SSR marker ranged from one to six with an average of three alleles per-locus (Table 3). The highest number of alleles was observed in the SSRBO14256 locus (six alleles) and the lowest number of alleles was observed in the SSRBO16864, SSRBO29, and SSRBO1064 loci (one allele) (Table 3). The effective number of alleles for each locus ranged from 1.00 to 2.58, with an average of 1.76. The allele size ranged from 105 bp (SSRBO1064) to 247 bp (SSRBO3955) (Table 3). The Shannon index (I) varied between zero in the SSRBO16864, SSRBO29, and SSRBO1064 loci to 1.075 in the SSRBO9629 locus. The expected heterozygosity ranged from 0.439 in the SSRBO14256 locus to 0.603 in the SSRBO9629 locus, averaging 0.331 for all loci. The observed heterozygosity ranged from 1.000 in the SSRBO8819 and SSRBO20205 loci to 0.519 in the SSRBO9629 locus, averaging 0.477 for all loci (Table 3). Most markers exhibited relatively high probabilities of null alleles, ranging from −0.280 to −0.104. In contrast, SSRBO9629 showed a notably lower value of 0.052. The PIC value varied from 0.00 (SSRBO16864, SSRBO29, and SSRBO1064) to 0.70 (SSRBO8819) with a mean value of 0.37 (Table 3).

Across the sampled populations, the number of different alleles (Na) ranged from 1.875 in the TOTO population to 2.375 in the QCA population, while the number of effective alleles (Ne) varied between 1.696 (TOTO) and 1.880 (QCA) (Table 4). The Shannon index (I) ranged from 0.489 in the TOTO population to 0.603 in the QCA population. The highest observed heterozygosity (Ho) was recorded in QSN (Ho = 0.500), while the lowest was found in QCA (Ho = 0.450) (Table 4). The highest expected heterozygosity (He) was observed in QCA (0.346), and the lowest was in TOTO (0.321). Additionally, the average observed heterozygosity was higher than the expected heterozygosity in all populations. The F value ranged from −0.528 to −0.313 across the populations (Table 4).

Regarding the genetic distance between populations, the lowest genetic distance was observed between the TOTO and QCA populations (0.059), while the highest genetic distance was found between the TOTO and QH populations (0.112) (Table 5).

The UPGMA dendrogram revealed that the four populations could be grouped into three main clusters (Figure 4). Cluster 1 includes a subset of the QH population. Cluster 2 comprises the TOTO and QCA populations, while cluster 3 consists of the QSN population along with portions of the TOTO and QH populations (Figure 4).

All *B. ovallei* individuals were analyzed using STRUCTURE to infer their population genetic structure. The optimal number of clusters (K) varied depending on the method used. Pritchard’s method identified K = 3 as the best fit (Figure 5A), while Evanno’s ΔK method suggested K = 5 (Figure 5B). The genetic composition of individuals showed a high degree of admixture across all populations (Figure 5C). In this study, K = 3, as determined by Pritchard’s method, was chosen as the most appropriate number of clusters.

## 3. Discussion

The genetic diversity of the plants in the Desierto Florido of the Atacama region is of significant ecological and conservational interest [2]. The Lion’s Claw species only emerges when rainfall is sufficient, typically requiring around 30–50 mm of precipitation. However, such rainfall is rare, occurring only every 3 to 7 years, which makes this species infrequently observed. Undoubtedly, rainfall variability plays a crucial role in determining which species can bloom and reproduce during each Desierto Florido event [3]. Furthermore, habitat fragmentation can influence gene flow between populations, affecting genetic diversity through processes of genetic exchange and isolation [19].

### 3.1. Novel SSR Marker for B. ovallei Using NGS

The use of next-generation sequencing (NGS) technology has revolutionized the identification and development of microsatellite markers (SSRs, simple sequence repeats), providing more efficient and accurate tools for the study of genetic diversity in plants [20]. In this study, we randomly obtained sequences base on de novo genome assembly from the nuclear genome of *B. ovallei*, being sufficient for the development of 268 microsatellite loci. To date, we have not found any published studies on genetic diversity in populations of species from the Alstroemeriaceae family using microsatellite markers. It is important to note that co-dominant markers provide more genetic information than dominant markers, as they allow the identification of both homozygous and heterozygous individuals [12].

In our study, we detected a higher number of trinucleotide repeats in *B. ovallei*, followed by di-, tetra-, hexa-, mono-, and pentanucleotide repeats. In contrast, most published studies that have developed microsatellite markers for plant species report a predominance of mononucleotide repeats, followed by di-, tri-, tetra-, penta-, and hexanucleotide repeats [21,22,23]. This difference is likely due to the parameters used in those studies, which typically set a minimum threshold of six repeats for dinucleotides and four or five for mononucleotides and other repeat types. In our case, we applied different parameters, requiring a minimum of 20 repeats for mononucleotides, 10 for dinucleotides, and five for all other repeat types.

This study presents, for the first time, 34 co-dominant microsatellite markers for *B. ovallei*, which can be transferred to other species of the genus *Bomarea* for genetic diversity studies, among other applications. In species of the *Alstroemeriaceae* family, the earliest studies on interspecific genetic diversity used dominant markers such as RAPD [24,25]. Regarding co-dominant markers, a genetic diversity and phylogenetic study used AFLP markers to evaluate the relationships between Chilean and Brazilian Alstroemeria species [26]. More recently, a comprehensive phylogenomic study employing hundreds of nuclear loci was published to infer macroevolutionary patterns and understand the diversification and biogeographic history of the genus *Bomarea* [8]. However, studies on intraspecific genetic variability or population-level diversity in *Bomarea* remain scarce.

### 3.2. Genetic Diversity Assessment of B. ovallei

In this study, we conducted the first genetic diversity analysis of four populations of the endemic and endangered species *B. ovallei* using a set of eight fluorescently labeled primer pairs. In our analysis, five SSR markers (SSRBO8819, SSRBO9629, SSRBO14256, and SSRBO20205) exhibited high PIC values (ranging from 0.48 to 0.70), suggesting a high level of genetic information among the 39 *B. ovallei* individuals. However, three SSR markers (SSRBO16864, SSRBO29, and SSRBO1064) were not polymorphic.

In general, the average genetic diversity values of 39 *B. ovallei* individuals (Na = 3 and He = 0.331) were low compared to values found in species of the same order as *Bomarea* (*Liliales*), such as the *Lilium longiflorum* population from Ishigakijima Island (Na = 10.3 and He = 0.745) [27], *Tricyrtis macropoda* (Na = 5.25 and He = 0.578), *Tricyrtis setouchiensis* (Na = 6.13 and He = 0.703), and *Tricyrtis affinis* (Na = 4.5 and He = 0.555) [28], using a similar number of individuals and microsatellite markers (between eight and 10 loci).

The same study on *Lilium longiflorum* Thunb. analyzing two populations found that the population on Yakushima Island exhibited the lowest genetic diversity, with values of Na = 3.2 and He = 0.346 [27], which predominantly consisted of self-compatible individuals [29]. Similarly, *B. ovallei* showed low genetic diversity, comparable to the *L. longiflorum* population from Yakushima Island. Furthermore, the same study found that in the Yakushima population, three out of ten loci displayed only a single detected allele, a pattern also observed in *B. ovallei*, where three loci showed only one allele (SSRBO16864, SSRBO29, and SSRBO1064). These comparisons help to confirm that the *B. ovallei* populations exhibit low genetic diversity.

The self-compatible mating system in a species leads to lower genetic diversity, as observed in *L. longiflorum* [27]. Regarding the mating system of *Bomarea* species, no published studies are available; therefore, more specific research is needed. Similarly, the mating system of *Alstroemeria aurea* Graham (*Alstroemeriaceae*) presents self-compatibility and exhibits synchronous dichogamy, meaning that at different times, an entire inflorescence can function as male, neuter, or female [30]. Additionally, restricted gene flow has been observed in this species [30,31]. Inbreeding depression has been documented in *A. aurea* as a consequence of self-pollination, resulting in reduced seed production and smaller seed size [31]. Similarly, in *Cariniana legalis* Mart., inbreeding depression has been found to be more pronounced in selfed seedlings than in those resulting from crosses between related individuals [32]. The low genetic diversity observed in *B. ovallei* populations probably may indicate that this species could use a self-compatible mating system or mixed mating system. Moreover, species endemism and the limited distribution of populations, such as *B. ovallei*, could contribute to reduced genetic diversity. For example, in a study that used AFLP markers, among eight closely related species of the genus *Fritillaria* L. (*Liliaceae*; *Liliales*), three endemic species (*F. yuzhongensis*, *F. dajinensis*, and *F. sinica*) showed relatively low genetic diversity with HPOP of 0.1935, 0.1785, and 0.2088, respectively [33]. On the other hand, in our study, most markers showed a relatively high probability of null alleles, which may reflect the effects of inbreeding. Inbreeding naturally reduces observed heterozygosity (Ho), as related individuals are more likely to share the same alleles, increasing homozygosity within the population [34].

Regarding the genetic diversity level among *B. ovallei* populations, genetic diversity was highest in the QCA population (Na = 2.375; I = 0.603; He = 0.346), followed by the QSN (Na = 2.250; I = 0.554; He = 0.334), QH (Na = 2.125; I = 0.531; He = 0.324), and TOTO (Na = 1.875; I = 0.489; He = 0.321) populations. The lower genetic diversity in the TOTO population is likely due to its proximity to the town of Totoral, where multiple anthropogenic activities take place daily, such as tourism, proximity to the main road axis, access to secondary roads, potential mining activities, etc. However, the QSN and QH populations are closer to the marine coast, where there are fewer road access points and the terrain is more rugged, making it more difficult for people to reach these sites. Additionally, the QCA population is located within Llanos de Challe National Park, where individuals are more protected than in the other populations. Undoubtedly, the low genetic diversity in the TOTO population is a warning sign, highlighting the need for future efforts to protect this population, which is highly exposed to anthropogenic activities. Furthermore, it underscores the importance of expanding the National Park’s management to include this area.

### 3.3. Genetic Relationships and Population Structure

The genetic relationships among the 39 *B. ovallei* individuals, as revealed by the UPGMA dendrogram, indicate a complex pattern of genetic clustering across the four studied populations. The formation of three main clusters suggests a partial genetic differentiation among populations, although some overlap is evident. Cluster 1, which includes only a subset of the QH population, may reflect a more genetically distinct subgroup within this location. Cluster 2, composed of individuals from both TOTO and QCA, implies a close genetic relationship or potential gene flow between these two populations. Cluster 3, which includes individuals from QSN as well as portions of the TOTO and QH populations, further supports the presence of admixture and gene flow among geographically distinct sites.

These findings are consistent with the results of the STRUCTURE analysis, which revealed a high level of admixture among individuals from all populations. Although different methods yielded different optimal numbers of genetic clusters—Pritchard’s method suggesting K = 3 and Evanno’s ΔK method indicating K = 5—this discrepancy is not uncommon in population genetics studies. In this case, K = 3 was considered the most appropriate due to the balance it offers between biological interpretability and statistical support.

The high degree of admixture observed suggests that despite geographic separation, there has been historical or ongoing gene flow among populations of *B. ovallei*. The lack of strong population structure may also reflect relatively recent divergence or the influence of human-mediated disturbances on gene flow dynamics.

Overall, the integration of clustering (UPGMA) and Bayesian assignment (STRUCTURE) approaches highlights the complex genetic landscape of *B. ovallei*, emphasizing the importance of using multiple methods to understand population structure. These insights are crucial for conservation planning, particularly for the maintenance of genetic diversity and connectivity among populations.

### 3.4. Practical Implications

Some mechanistic insights into *B. ovallei*, particularly concerning gene function, may be associated with the SSR loci identified in this study. Although SSR markers are typically considered neutral, it is plausible that certain loci may be linked to genomic regions or traits with adaptive significance—especially in a species that has survived the extreme environmental conditions of the Atacama Desert [35]. Comparative analyses with closely related species within the *Alstroemeriaceae* family could further illuminate the evolutionary relevance of these loci.

In terms of practical breeding implications, the validated SSR markers developed in this study hold considerable potential for applications in conservation and selective breeding. These include genetic diversity monitoring, germplasm characterization, and marker-assisted selection. Encouragingly, viable rhizomes of *B. ovallei* can already be obtained through the transplantation of seedlings from in vitro to ex vitro conditions, offering a feasible approach for propagation and ex situ conservation efforts [36].

## 4. Materials and Methods

### 4.1. Materials

From August to September 2022, the Desierto Florido phenomenon emerged in the Atacama Region, with a significant number of Lion’s Claw (*Bomarea ovallei*) individuals observed across various sectors of Llanos de Challe National Park and its surrounding areas, including both the coastal zone and the vicinity of the town of Totoral. Under a permit granted by CONAF (authorization no. 87/2022, 21 October 2022), due to the species’ conservation status, fresh leaves were collected of 39 individuals of *B. ovallei* with red flowers, from four populations: Totoral (TOTO), Quebrada Honda (QH), Quebrada Sin Nombre (QSN), and Quebrada Carrizal (QCA) (Table 6). No more individuals were collected in each area to avoid harming the few plants that appeared. Specifically, one leaf per individual was collected, stored in a cooler at 9 °C, and later the same day transferred to a laboratory ultrafreezer at −80 °C for preservation. The taxonomic identification of the species was conducted based on the descriptions provided by Ravenna [37]. Table 6 provides the locations of the georeferenced samples. One individual was processed for herbarium preservation and deposited at the Herbarium (EIF, Index Herbariorum Code) of the Universidad de Chile, Department of Forestry and Nature Conservation, under the voucher number 17379.

### 4.2. DNA Extraction and NGS Technology

DNA was extracted from the leaf samples using a modified cetyl-trimethylammonium bromide (CTAB) protocol, as outlined by Contreras et al. [38]. The quality and concentration of the genomic DNA extracted from the 39 samples were assessed using a Colibri microvolume spectrophotometer (Titertek-Berthold, Pforzheim, Germany). The DNA extracted from one individual of *B. ovallei* was quantified with a Qubit™ 3.0 fluorometer and a Qubit™ dsDNA HS Assay Kit (Life Technologies, San Diego, CA, USA), following the manufacturer’s instructions. The DNA samples from *B. ovallei* were stored at −80 °C, and their integrity was verified using an Agilent 2100 Bioanalyzer (Agilent Technologies, San Diego, CA, USA) prior to sequencing. Sequencing libraries were prepared using a TruSeq Nano DNA LT Kit (Illumina, San Diego, CA, USA). The final libraries were analyzed on the Agilent 2100 Bioanalyzer to confirm fragment size distribution and concentration. Sequencing was conducted at Novogene Inc. (Sacramento, CA, USA) using an Illumina sequencing platform.

Raw sequencing reads underwent a rigorous filtering process. Reads containing >10% of bases with a quality score below Q30, as well as those representing noncoding RNA, ambiguous sequences, empty reads, and adaptor contaminants, were removed. To ensure the accuracy and reliability of the SSR search, contigs shorter than 300 bp were filtered out and excluded. The forward and reverse raw sequences were merged using PEAR version 0.9.4 [39]. The sequencing data have been submitted to the National Center for Biotechnology Information (NCBI).

### 4.3. SSR Locus Search

SSR markers were identified across the assembled genome using MegaSSR [40]. The search targeted SSR motifs ranging from mono- to hexanucleotides, with the following minimum repeat unit thresholds: 20 repeat units for mononucleotides, 10 for dinucleotides, and 5 for tri-, tetra-, penta-, and hexanucleotides. Primer pairs for the selected SSR loci were designed using Primer3 software [41]. The design parameters included a preferred amplicon size of 100–350 bp, primer size of 18–27 bp, a primer melting temperature of 55–60 °C, and a GC percentage between 30 and 70%.

### 4.4. Evaluation of New SSR Markers

A total of 34 primer pairs were randomly selected and synthesized for *B. ovallei*. These primers were then evaluated for their presence or alignment within the chloroplast genome and the incomplete mitogenome sequences of the *B. ovallei*, with no matches found in these organelles. PCR was conducted in a total volume of 24 µL, comprising 12 µL of SapphireAmp Fast PCR 2X Master Mix (Takara-Clontech, San Jose, CA, USA), 3 µL of genomic DNA (5 ng/µL), 1.0 µL of each primer (forward and reverse) at a concentration of 5 µM, and 7 µL of nuclease-free water. PCR amplification was performed in a Labnet MultiGene OptiMax Thermal Cycler using the following protocol: initial denaturation at 94 °C for 3 min; 40 cycles of 94 °C for 25 s, Ta °C (see Table 2 and Appendix A) for 25 s, and 72 °C for 40 s; followed by a final extension at 72 °C for 3 min. The PCR products were analyzed by electrophoresis on 2.0% agarose gels stained with GelRed DNA stain (10,000X, Biotium Inc., Fremont, CA, USA). Band sizes were estimated using a 100 bp DNA ladder (Thermo Fisher, Foster City, CA, USA). A subset of eight randomly selected primer pairs that produced strong bands in a preliminary test with 39 individuals were chosen for further analysis using fragment analysis with fluorophore-labeled primers.

Fluorophore-labeled primer pairs were tested on ten *B. ovallei* individuals from each of the four populations. PCR was carried out in a total volume of 50 µL, consisting of 25 µL of SapphireAmp Fast PCR 2X Master Mix (Takara-Clontech, San Jose, CA, USA), 4 µL of genomic DNA (5 ng/µL), 0.8 µL of each primer (forward primer labeled with FAM or HEX fluorophores) at a concentration of 5 µM, and 17 µL of nuclease-free water. The amplification program followed the same protocol as described earlier. PCR products were analyzed using an ABI3730XL Genetic Analyzer (Applied Biosystems, Foster City, CA, USA), and allele sizes were determined with Peak Scanner Software (Applied Biosystems, version 1.0) using the GeneScan 400 HD ROX size standard (Applied Biosystems).

### 4.5. Data Analyses

Genotyping errors in microsatellite data for each locus were assessed using MICRO-CHECKER v.2.2.3 [42]. GenAlex 6.5 software [43] was used to characterize each SSR locus and assess genetic diversity across populations. The analyzed parameters included the number of alleles per locus (Na), the number of effective alleles (Ne), the Shannon information index (I), observed heterozygosity (Ho), expected heterozygosity (He), fixation index (F), and Nei’s genetic distance. The probability of null alleles was estimated using the formula described in [44]: r = (He − Ho)/(1 + He). The polymorphism information content (PIC) for each SSR locus was calculated using the formula PIC = 1 − Σp^2^, where *p* represents the frequency of different alleles detected at a given locus. A PIC value below 0.25 indicates low polymorphism, a value between 0.25 and 0.5 indicates moderate polymorphism, and a value greater than 0.5 indicates a highly polymorphic locus [45].

Cluster analysis of 39 individuals was conducted using the unweighted pair-group method with arithmetic mean (UPGMA) algorithm based on the Euclidean similarity index. A dendrogram was created using PAST software version 3.22 [46].

Population genetic structure was analyzed using a Bayesian clustering approach implemented in STRUCTURE v2.3.4 [47]. Ten independent runs were conducted for each K value, with a run length of 100,000 Markov Chain Monte Carlo (MCMC) iterations following a burn-in period of 100,000 iterations. The analysis assumed an admixture model with correlated allele frequencies. To identify the most likely number of genetic clusters (K), both the method based on the log-likelihood of the data, LnP(K), proposed by Pritchard et al. [47], and the ΔK method of Evanno et al. [48], were applied. The results were calculated using the web-based tool StructureSelector [49], which implements both methods.

## 5. Conclusions

This study represents the first genetic diversity assessment of *Bomarea ovallei* using microsatellite markers derived from de novo genome assembly. A total of 268 SSR loci were identified, with trinucleotide repeats being the most prevalent. From these, 34 primer pairs were developed, and eight were used to analyze 39 individuals from four populations. The results revealed low overall genetic diversity (Na = 3; He = 0.331), with three loci showing no polymorphism. Notably, the QCA population exhibited the highest diversity, while the TOTO population showed the lowest—likely due to anthropogenic pressures near urbanized areas. STRUCTURE and UPGMA analyses indicated complex genetic clustering and high admixture, suggesting gene flow between populations despite geographic barriers. The observed low genetic diversity may result from limited distribution, potential self-compatibility, and habitat fragmentation. These findings underscore the importance of conserving the TOTO population and expanding protective management to include surrounding areas. Additionally, the development of transferable co-dominant SSR markers provides valuable tools for future population genetics studies within the *Bomarea* genus. The integration of next-generation sequencing, genetic diversity metrics, and clustering approaches highlights the need for multi-faceted methods in conservation genetics, especially for endemic species in fragile ecosystems such as the Desierto Florido. This research contributes critical baseline data for guiding conservation strategies aimed at preserving genetic diversity and population connectivity in *B. ovallei*.

## Figures and Tables

**Figure 1 plants-14-01468-f001:**
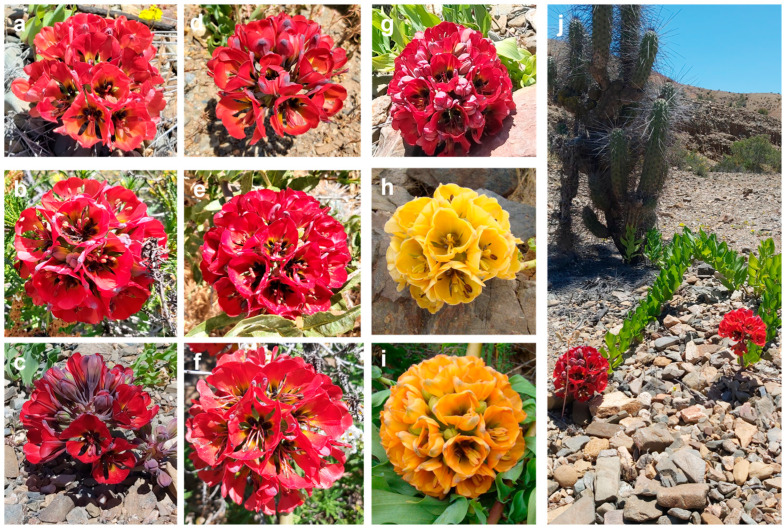
Variation in flower head color, predominantly red (**a**–**i**), and the creeping growth form of *Bomarea ovallei* (**j**).

**Figure 2 plants-14-01468-f002:**
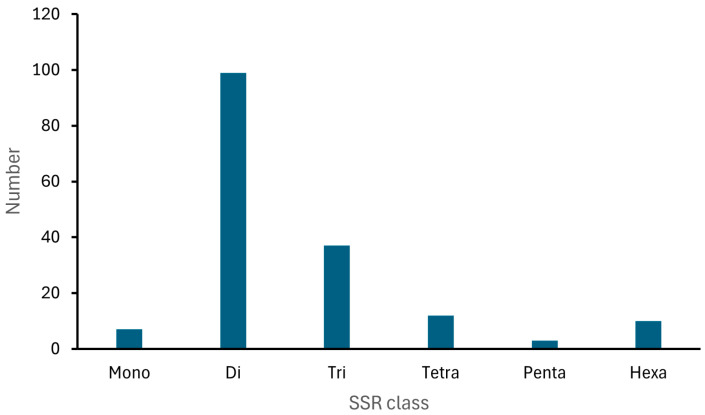
Distribution of the different SSR classes in *Bomarea ovallei*.

**Figure 3 plants-14-01468-f003:**
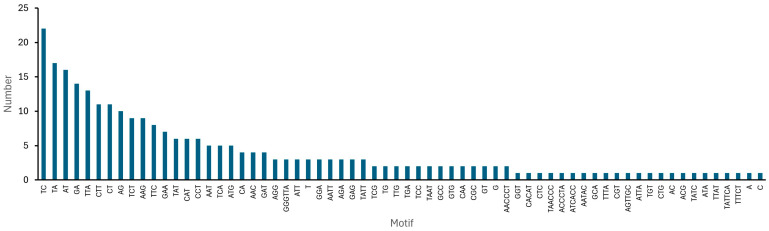
Frequency of the identified SSR motifs in *Bomarea ovallei*.

**Figure 4 plants-14-01468-f004:**
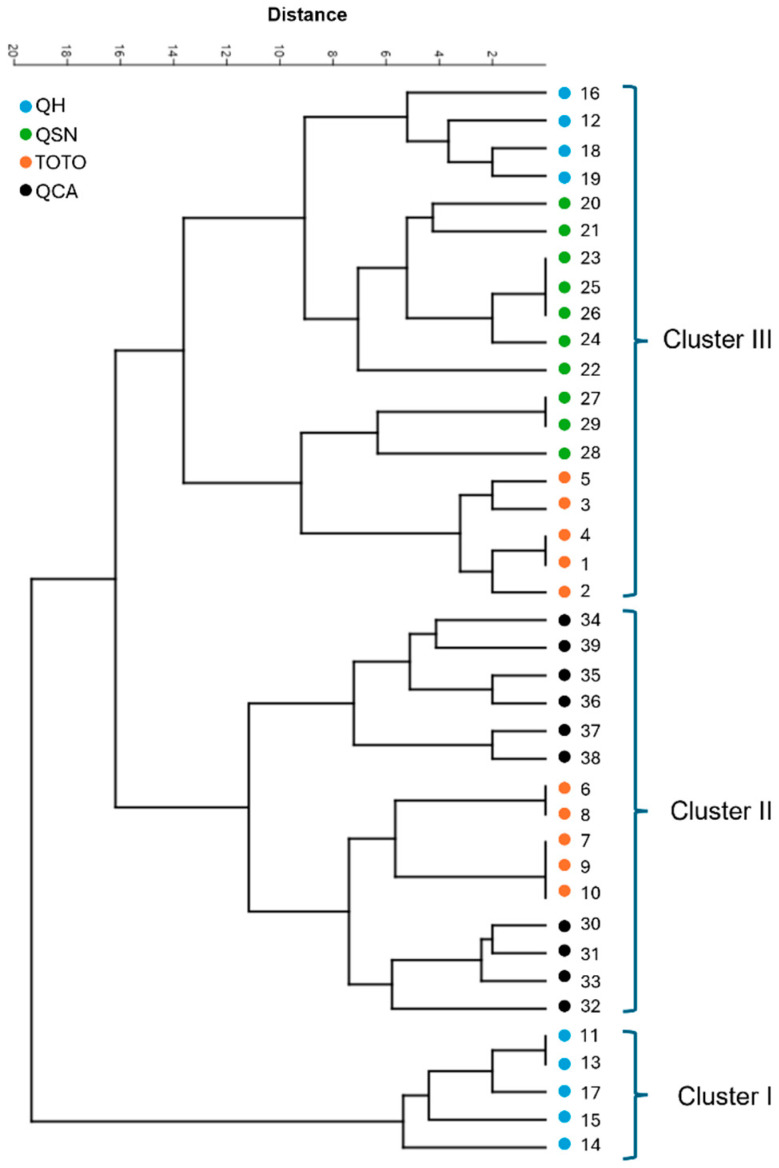
UPGMA dendrogram of 39 *Bomarea ovallei* individuals based on eight SSR markers developed in this study. Brackets on the left denote three main clades. The scale bar indicates genetic distance.

**Figure 5 plants-14-01468-f005:**
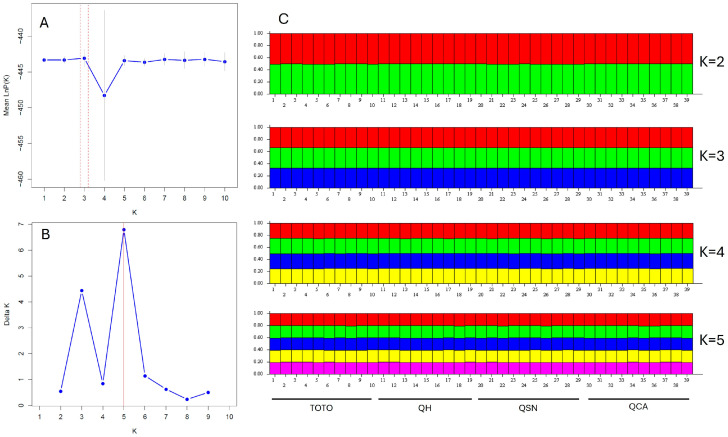
(**A**) Average log-likelihood values [lnP(K)] with corresponding standard deviations of posterior probabilities. (**B**) Delta K values plotted for K ranging from 1 to 10, produced using StructureSelector. (**C**) STRUCTURE-inferred genetic structure bar plot of four populations, based on the SSR marker data.

**Table 1 plants-14-01468-t001:** Results of microsatellite search from *Bomarea ovallei* using MegaSSR software.

Category	Total Number
Total number of sequences examined	18,709
Total size examined sequences (bp)	6,252,133
Total number of identified SSRs	268
Number of SSR-containing sequences	254
Number of sequences containing more than one SSR	13
Number of SSRs present in compound formation	9

**Table 2 plants-14-01468-t002:** Characteristics of a novel set of eight SSR loci developed for *Bomarea ovallei*.

Locus	5′-Sequence-3′	Accession	Motif	Size (bp)	T°a (°C)
SSRBO8819	F-AGTCGAGTTTCTCTGATCTTAAA R-ACAGAATCCAAAATAAAACTGGG	PV126292	(TTA)_9_	153	59
SSRBO16864	F-ATCATCAAAGATAGTAGCGGAAT R-CTGATCTTTGTCTCTCTCCG	PV126301	(TCA)_9_	169	59
SSRBO9629	F-CCCTTTTATGGTTTGTTGTAGCA R-AGCCAAGAACAAACCGTAAAG	PV126315	(CT)_21_	195	59
SSRBO14256	F-GGTCTTAGGGTGTGGGCTAA R-GTGGCATGGAAAGGAACACA	PV126317	(CT)_14_	169	59
SSRBO29	F-TGTTCAATCAAGTCATGGGCA R-CTGGTGTAGTATCATGCATGCA	PV126319	(TGA)_7_	204	59
SSRBO3955	F-TCGCACTTTCTCTCTGTCCT R-AAACTCTATCTAAACATCCGCCA	PV126320	(TGA)_6_	232	59
SSRBO1064	F-TGAGTTGCTGTGGTGGTATG R-TCACCAAACAACCATAACACCA	PV126321	(TTG)_11_	105	57
SSRBO20205	F-TCAGGCGATTGGTTGGAAAG R-GTTTGGGACGCGGTTTCTTT	PV126324	(TTC)_7_	118	59

**Table 3 plants-14-01468-t003:** Characteristics of the eight fluorescently labeled SSR primer pairs used in this study.

Locus	Allele Size (bp)	Na	Ne	I	Ho	He	Null Allele	PIC
SSRBO8819	144, 147, 150, 156	4	2.54 (0.21)	0.991 (0.11)	1.000 (0.000)	0.598 (0.035)	−0.251	0.70
SSRBO16864	154	1	1.00	0	0	0	-	0.00
SSRBO9629	197, 199, 201, 203	4	2.58 (0.24)	1.075 (0.07)	0.519 (0.089)	0.603 (0.036)	0.052	0.69
SSRBO14256	167, 169, 171, 173, 181, 183	6	1.80 (0.10)	0.715 (0.08)	0.589 (0.042)	0.439 (0.032)	−0.104	0.48
SSRBO29	195	1	1.00	0	0	0	-	0.00
SSRBO3955	229, 235, 241, 247	4	1.92 (0.28)	0.691 (0.11)	0.711 (0.167)	0.448 (0.071)	−0.181	0.53
SSRBO1064	105	1	1.00	0	0	0	-	0.00
SSRBO20205	109, 112, 115	3	2.29 (0.09)	0.884 (0.06)	1.000 (0.000)	0.562 (0.021)	−0.280	0.56
Average		3	1.76 (0.12)	0.545 (0.08)	0.477 (0.075)	0.331 (0.048)		0.37

Note: total number of alleles per locus (Na), effective number of alleles (Ne), Shannon’s information index (I), observed heterozygosity (Ho), expected heterozygosity (He), probability of null alleles, and polymorphic information content (PIC). Values in parentheses are standard deviation (±).

**Table 4 plants-14-01468-t004:** Genetic diversity indices in the four *Bomarea ovallei* populations based on eight SSR loci.

Population	N	Na	Ne	I	Ho	He	Fis
TOTO	10	1.875 (0.29)	1.696 (0.23)	0.489 (0.11)	0.488 (0.164)	0.321 (0.097)	−0.528 (0.190)
QH	9	2.125 (0.39)	1.752 (0.26)	0.531 (0.17)	0.472 (0.154)	0.324 (0.102)	−0.441 (0.073)
QSN	10	2.250 (0.41)	1.736 (0.22)	0.554 (0.16)	0.500 (0.165)	0.334 (0.099)	−0.486 (0.169)
QCA	10	2.375 (0.46)	1.880 (0.31)	0.603 (0.19)	0.450 (0.151)	0.346 (0.109)	−0.313 (0.131)

Note: samples number (N), number of different alleles (Na), effective number of alleles (Ne), Shannon’s information index (I), observed heterozygosity (Ho), expected heterozygosity (He), inbreeding coefficient (Fis). Values in parentheses are standard deviation (±).

**Table 5 plants-14-01468-t005:** Pairwise Nei’s genetic distance values between populations of *Bomarea ovallei*.

	TOTO	QH	QSN	QCA
TOTO	0.000			
QH	0.112	0.000		
QSN	0.063	0.103	0.000	
QCA	0.059	0.078	0.092	0.000

**Table 6 plants-14-01468-t006:** Georeferencing of 39 individuals and populations of Lion’s Claw collected in this study.

N°	Latitude S	Longitude W	Altitude (Masl)	Populations
1–10	27°53′45.09″ S	70°59′17.25″ W	153–162	TOTO
11–19	27°59′25.39″ S	71°7′49.86″ W	83–93	QH
20–29	28°01′39.4″ S	71°07′51.2″ W	84–91	QSN
30–39	28°6′27.64″ S	71°6′23.81″ W	39–41	QCA

## Data Availability

Data is contained within the article and Appendix A; further inquiries can be directed to the corresponding author.

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
