# Peer review of "Genetic Diversity and Conservation of Bomarea ovallei (Phil.) Ravenna: Microsatellite Markers Reveal Population Vulnerability in the Atacama Desert"

_plants, 2025, doi:10.3390/plants14101468_

Round 1
Reviewer 1 Report
Comments and Suggestions for Authors
I have completed the review of the manuscript, and I find it to be well-written and suitable for publication. The study provides valuable insights into the genetic diversity of Bomarea ovallei and offers important contributions to conservation genetics.
My comments and suggestions are included in the attached file. Please let me know if any further clarifications are needed.

Author Response
Comments Reviewer 1.
We thank the reviewer for their valuable comments and address them in detail below.
Line 2. Response. Thank you for your comment. Correct the scientific name as suggested.
Lines 87-89. Response. The suggested manuscript provides a valuable example of genetic diversity patterns in small populations; therefore, we have included it in our revised manuscript.
Line 128. Table 3. I suggest including this table as supplementary material. Response. We believe it is important to retain Table 3 in the main body of the manuscript, as this represents the first report of SSR markers developed for the genus Bomarea. These newly developed markers provide a valuable resource for future studies and can be tested for cross-species transferability across the more than 120 recognized species within the genus.
Line 130. Response. Scientific name corrected to italics
Line 144. Table 4. I would like to see a test for null alleles in these markers. Response. We have added in a column the probability of null alleles for each marker.
Line 159. Table 5. Is this Fis? Could you please show the Fst values for pairs of these populations?. Response. Indeed, the "F" refers to Fis, and we have clarified this to improve understanding. We chose not to include the Fst value because the main objective of the manuscript is to assess genetic diversity within populations, rather than to evaluate genetic differentiation among populations. In any case, the overall Fst value was 0.107, indicating a moderate level of genetic differentiation among populations.
Line 210. Please add a reference to support this statement. Response. Thank you for your comment. We have added the reference that supports the statement.
Lines 242 and 245. Response. We have corrected the scientific name.
Lines 257-258. Add new reference. Response. The new reference has been added, and we believe it provides additional support for the assertion.
Line 260. Response. we agree to add the phrase suggested by the referee.
Line 369. Response. Ta °C was corrected.
Line 393. Response. We have corrected the formula and eliminated the letter “i”.
Reviewer 2 Report
Comments and Suggestions for Authors
-
For Figure 1, each sub-figure should be labeled with letters (a, b, c, d, e, f, g, h).
-
SSR markers can be broadly classified into two categories: conventional SSRs visualized by silver staining after gel electrophoresis, and fluorescence-labeled SSRs. The authors are suggested to include this information in the Introduction section. Additionally, the rationale for selecting SSR markers over other available molecular markers should be explicitly discussed.
-
The Results section should specify the source of SSR screening data - whether obtained through next-generation sequencing (NGS) in this study or derived from existing literature.
-
In Figure 2, the width of the histogram bars appears unnecessarily wide and should be adjusted.
-
For Table 3 listing SSR primers, it should be clarified whether all primers were newly designed in this study or some were adopted from previous publications.
-
Latin names should be properly formatted throughout the manuscript, with genus and species names italicized (e.g., Bomarea ovallei).
-
The discrepancy between Table 3 (listing 24 SSR primer pairs) and Table 4 (including only 8 pairs) requires explanation.
-
For Figure 4 displaying clustering results, it is recommended to correlate the materials with flower color variations, particularly if this study focuses on SSR analysis of flower color polymorphism in Bomarea ovallei. Furthermore, since flower color shows significant correlation with environmental factors, particularly temperature and humidity, these relationships should be considered in the analysis.
-
The Discussion section would benefit from being organized into subsections to enhance clarity and logical presentation of key findings.
-
The reference list should be updated to include recent publications, particularly those from 2024 and 2025, to reflect current research progress.
- To strengthen the study, consider supplementing with additional data: Morphological traits (flower + leaf characteristics), Physiological and biochemical indicators, Cytogenetic data (FISH chromosome analysis). This comprehensive approach would better demonstrate population differentiation in Bomarea ovallei.
Author Response
Comments Reviewer 2.
We thank the reviewer for their valuable comments and address them in detail below.
1.- For Figure 1, each sub-figure should be labeled with letters (a, b, c, d, e, f, g, h).
Response: Thank you for your comment. We have labeled each subfigure with the letters: a, b, c, d, e, f, g, h, i and j.
2.-SSR markers can be broadly classified into two categories: conventional SSRs visualized by silver staining after gel electrophoresis, and fluorescence-labeled SSRs. The authors are suggested to include this information in the Introduction section. Additionally, the rationale for selecting SSR markers over other available molecular markers should be explicitly discussed.
Response: Thank you for your insightful comment. In response, we have revised the Introduction to include a brief explanation of the two main categories of SSR markers: conventional SSRs typically visualized through silver staining following gel electrophoresis, and fluorescence-labeled SSRs used in capillary electrophoresis platforms (MS, lines 80-82). This distinction helps clarify the methodological framework of our study and aligns with the fragment analysis approach we implemented using fluorophore-labeled primers.
We have also added a justification for the selection of SSR markers over other types of molecular markers. Specifically, we note that SSRs were chosen due to their co-dominant inheritance, high polymorphism, reproducibility, and broad transferability across related species. These features make them particularly suitable for assessing genetic diversity and population structure in non-model and conservation-priority species such as Bomarea ovallei, especially when sample size is limited and population connectivity is of concern. These statement can be found in the revised Introduction (MS, lines 85-89).
3.-The Results section should specify the source of SSR screening data - whether obtained through next-generation sequencing (NGS) in this study or derived from existing literature.
Response: Thank you for your comment. As suggested, we have specified this in the Results section, line 168, as follows: 'A total of 268 microsatellite loci were identified de novo from the assembled contigs generated via next-generation sequencing (NGS), utilizing the MegaSSR software.'"
4.-In Figure 2, the width of the histogram bars appears unnecessarily wide and should be adjusted.
Response: In accordance with the reviewer’s suggestion, we have adjusted Figure 2 to display narrower bars.
5.-For Table 3 listing SSR primers, it should be clarified whether all primers were newly designed in this study or some were adopted from previous publications.
Response: This study represents the first release of a set of SSR markers developed for species within the genus Bomarea. The title of Table 3 was modified as follows: “Characteristics of a novel set of 34 SSRs loci in Bomarea ovallei”
6.-Latin names should be properly formatted throughout the manuscript, with genus and species names italicized (e.g., Bomarea ovallei).
Response: We regret the omission of italics for some scientific names or genera and will review the manuscript thoroughly to avoid further errors.
7.-The discrepancy between Table 3 (listing 24 SSR primer pairs) and Table 4 (including only 8 pairs) requires explanation.
Response: Table 3 lists 34 primer pairs obtained from contigs through next-generation sequencing (NGS), while Table 4 presents eight fluorescently labeled SSR primer pairs. To enhance clarity, we have revised the title of Table 4 to: 'Characteristics of the eight fluorescently labeled SSR primer pairs used in this study.
8.-For Figure 4 displaying clustering results, it is recommended to correlate the materials with flower color variations, particularly if this study focuses on SSR analysis of flower color polymorphism in Bomarea ovallei. Furthermore, since flower color shows significant correlation with environmental factors, particularly temperature and humidity, these relationships should be considered in the analysis.
Response: Although flower coloration is an interesting subject of study, it likely results from a complex interaction between genetic and biochemical factors. In our case, we sampled only individuals with red flowers, as those with yellow or orange flowers are less common. Consequently, this study focuses on the genetic diversity of red-flowered individuals across different populations. As stated in the Materials and Methods section (lines 329–330), we collected 39 red-flowered individuals from various populations.
9.-The Discussion section would benefit from being organized into subsections to enhance clarity and logical presentation of key findings.
Response: As recommended by the reviewer, we have reorganized the Discussion section into clearly defined subsections to enhance clarity and readability.
10.-The reference list should be updated to include recent publications, particularly those from 2024 and 2025, to reflect current research progress.
Response: We appreciate the reviewer’s suggestion regarding the inclusion of more recent references. However, after an extensive literature search, we found that there are very few publications from 2024 or 2025 related specifically to Bomarea ovallei or the genetic diversity of the Bomarea genus. To our knowledge, studies addressing these topics remain scarce, which further highlights the novelty and relevance of our work. Nonetheless, we have ensured that the reference list includes the most up-to-date and relevant literature currently available.
11.- To strengthen the study, consider supplementing with additional data: Morphological traits (flower + leaf characteristics), Physiological and biochemical indicators, Cytogenetic data (FISH chromosome analysis). This comprehensive approach would better demonstrate population differentiation in Bomarea ovallei.
Response: The morphological characterization of leaves and flowers was not possible, as the sampled individuals are no longer available. During the Desert Bloom event—which occurs infrequently—we were only permitted to collect a single leaf sample (5 cm²) per individual. Additionally, physiological and biochemical analyses could not be conducted due to restrictions outlined in the research permits, as this species is classified as endangered. Cytogenetic data for this species have already been reported in the following studies: https://doi.org/10.1007/s00606-021-01756-1 and https://doi.org/10.1080/00288250709509716.
Round 2
Reviewer 1 Report
Comments and Suggestions for Authors
After the suggestions, the authors significantly improved the manuscript and, therefore, it can be published in its current form.
Reviewer 2 Report
Comments and Suggestions for Authors
Reviewer's Comments on the Manuscript:
1) Figure 1:
The subfigure labels (a, b, c, d) in the lower-left corner should not necessarily use a white background, as it appears visually突兀 (jarring). Please adjust for better integration.
2) Table 3:
The table lists 34 pairs of SSR primers, but only 24 (or fewer) were actually effective in material identification. We recommend moving the full list of 34 primer pairs to a supplementary table and retaining only the validated primers in the main manuscript.
3) General Figures & Tables Formatting:
The overall layout of figures and tables requires improvement for better clarity and professionalism.
4) Figure 5B:
The top horizontal axis label is currently obscured. Please adjust the layout to ensure full visibility.
5) Additional Data Request:
The previous revision requested supplementary data to strengthen the study’s robustness (e.g., species-specific mechanistic insights or practical breeding implications). However, these critical additions were not incorporated in the current version. Without such enhancements, the manuscript remains insufficiently substantiated for journal publication standards.
Author Response
Comments 1. Figure 1:
The subfigure labels (a, b, c, d) in the lower-left corner should not necessarily use a white background, as it appears visually (jarring). Please adjust for better integration.
Response: Thank you for your valuable suggestion. We have revised the subfigure labels (a - j) to enhance their visual integration by removing the white background. The updated labels now harmonize more naturally with the figures while preserving clarity. We believe this adjustment improves the overall visual presentation, as recommended.
Comments 2. 2) Table 3:
The table lists 34 pairs of SSR primers, but only 24 (or fewer) were actually effective in material identification. We recommend moving the full list of 34 primer pairs to a supplementary table and retaining only the validated primers in the main manuscript.
Response: Thank you for your constructive feedback. Following your recommendation, we have relocated the complete list of 34 SSR primer pairs to a new supplementary table (Supplementary Table S1). The main manuscript now includes only the validated primers that proved effective for material identification. We believe this revision enhances the clarity and focus of the main text.
Comments 3. General Figures & Tables Formatting:
The overall layout of figures and tables requires improvement for better clarity and professionalism.
Response: We sincerely appreciate your constructive feedback. In response to your suggestion, we have undertaken a comprehensive revision of the layout of all figures and tables to enhance both clarity and professionalism. Specifically, we have standardized font sizes and styles, improved the alignment and spacing of elements, optimized figure resolutions, and refined table formatting to ensure ease of interpretation and a more polished presentation. We are confident that these revisions significantly enhance the overall quality and coherence of the manuscript.
Comments 4. Figure 5B:
The top horizontal axis label is currently obscured. Please adjust the layout to ensure full visibility.
Response: Thank you for bringing this issue to our attention. We have revised the figure layout to ensure full visibility of the top horizontal axis label. Specifically, we increased the top margin and adjusted the positioning of the axis title to prevent any overlap or cropping. These modifications enhance both the readability and overall presentation quality of the figure, in accordance with your recommendation.
Comments 5. Additional Data Request:
The previous revision requested supplementary data to strengthen the study’s robustness (e.g., species-specific mechanistic insights or practical breeding implications). However, these critical additions were not incorporated in the current version. Without such enhancements, the manuscript remains insufficiently substantiated for journal publication standards.
Response: Thank you for emphasizing the importance of reinforcing the study’s robustness with additional supporting information. In response, we have expanded the Discussion section to outline the practical implications of our findings. Specifically, we have added a detailed discussion (Lines 330–343) on the potential applications of the validated SSR markers in population monitoring, as well as in future breeding and conservation programs. We believe these additions enhance the relevance and applicability of the study, in accordance with your valuable recommendation.
